# Interactive Shape Based Brushing Technique for Trail Sets

Almoctar Hassoumi *
École Nationale de l'Aviation Civile

María-Jesús Lobo†
École Nationale de l'Aviation Civile

Gabriel Jarry ‡
École Nationale de l'Aviation Civile

Vsevolod Peysakhovich§
ISAE SUPAERO

Chrisophe Hurter ¶
École Nationale de l'Aviation Civile

## ABSTRACT

Brushing techniques have a long history with the first interactive selection tools appearing in the 1990s. Since then, many additional techniques have been developed to address selection accuracy, scalability and flexibility issues. Selection is especially difficult in large datasets where many visual items tangle and create overlapping. Existing techniques rely on trial and error combined with many view modifications such as panning, zooming, and selection refinements. For moving object analysis, recorded positions are connected into line segments forming trajectories and thus creating more occlusions and overplotting. As a solution for selection in cluttered views, this paper investigates a novel brushing technique which not only relies on the actual brushing location but also on the shape of the brushed area. The process can be described as follows. Firstly, the user brushes the region where trajectories of interest are visible (standard brushing technique). Secondly, the shape of the brushed area is used to select similar items. Thirdly, the user can adjust the degree of similarity to filter out the requested trajectories. This brushing technique encompasses two types of comparison metrics, the piecewise Pearson correlation and the similarity measurement based on information geometry. To show the efficiency of this novel brushing method, we apply it to concrete scenarios with datasets from air traffic control, eye tracking, and GPS trajectories.

**Index Terms:** Human-centered computing—Interaction design—Interaction design theory, concepts and paradigms——Human-centered computing—Visualization—Visualization application domainsVisual analytics Human-centered computing—Interaction design—Systems and tools for interaction design——Information systems—Information systems—Information retrievalInformation retrieval query processingQuery intent

## 1 INTRODUCTION

Brushing techniques [9], which are part of the standard InfoVis pipeline for data visualization and exploration [12], already have a long history. They are now standard interaction techniques in visualization systems [57] and toolkits [10, 50]. Such techniques help to visually select items of interest with interactive paradigms (i.e. lasso, boxes, brush) in a view. When the user visually detects a relevant pattern (i.e. a specific curve or a trend), the brushing technique can then be applied to select it. While this selection can be seamlessly performed, the user may still face issues when the view becomes cluttered with many tangled items. In such dense visualization, existing brushing techniques also select items in the vicinity of the target and thus capture part of the clutter (see Fig. 1). To address such an issue, the user can adjust the brushing parameters

---

*e-mail: almoctar.hassoumi-assoumana@isae.fr

†e-mail: maria-jesus.lobo@ign.fr

‡e-mail: gabriel.jarry@enac.fr

§e-mail: vsevolod.peysakhovich@isae.fr

¶e-mail: christophe.hurter@enac.fr

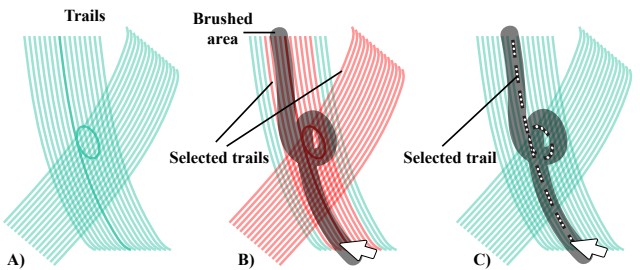

Figure 1: This figure shows our main rationale to define our shaped based brushing technique. A) Unselected trail set where the user wishes to select the only curved one. B) Standard brushing technique, where the brushing of the curved trail also selects every other trail which touches the brushing area. C) Our method uses the brushing input to compare with the brushed trajectory and only the trajectories similar in shape are selected.

by changing the brush size or the selection box locations. However, this may take time and requires many iterations or trials. This paper proposes a novel brushing technique that filters trajectories taking into account the shape of the brush in addition to the brush area. This dual input is provided at the same time and opens novel opportunities for brushing techniques. The cornerstone of such a technique relies on a shape comparison algorithm. This algorithm must provide a numerical similarity measurement which is ordered (low value for unrelated shapes, and high value for correlated shapes), continuous (no steps in the computed metric) and with a semantic so that the user can partially understand the logic behind this similarity measurement. Thus, to build such a dual filtering technique, the following design requirements (DR) must be fulfilled:

- DR1: The technique enables users to select occluded trajectories in dense or cluttered view.

- DR2: The shape comparison metric is flexible with continuous, ordered and meaningful values.

- DR3: The technique enables incremental selection refinement.

- DR4: The technique is interactive.

Taking into account the identified requirements (DR1–DR4), this paper presents a novel shape-based brushing tool. To the best of our knowledge, such a combination of brushing and shape comparison techniques has not yet been explored in trajectory analysis and this paper fills this gap. In the remainder of the paper, the following subjects are presented. First, previous works in the domain of brushing and shape comparison are provided. Second, the brushing pipeline is detailed with explanations of the comparison metric data processing. Next, the use of such a technique through different use-cases is demonstrated. The brushing technique is discussed in terms of usefulness, specific application and possible limitations. Finally, the paper concludes with a summary of our contribution and provides future research directions.

## 2 RELATED WORK

There are various domain-specific techniques targeting trail exploration and analysis. In this section, we explore three major components of selection techniques for trail-set exploration and analysis relevant to our work: brushing, query-by-content, and similarity measurement.

### 2.1 Brushing in Trajectory Visualization

Trail-set exploration relies on pattern discovery [13] where relevant trails need to be selected for further analysis. Brushing is a selection technique for information visualization, where the user interactively highlights a subset of the data by defining an area of interest. This technique has been shown to be a powerful and generic interaction technique for information retrieval [9]. The selection can be further refined using interactive filtering techniques [23]. The approach presented in this paper is based on dynamic queries [53] and direct manipulation [36, 52].

Systems designed specifically for spatio-temporal visualization and in particular in trajectory visualizations are very complex because of their 3D and time varying nature. Due to this, several systems and frameworks have been especially designed to visualize them [5, 6, 8, 32, 33, 51]. Most of these systems include selection techniques based on brushing, and some of them enable further query refinement through boolean operations [32, 33].

These techniques do not take into account the shape of the trails, so selecting a specific one with a particular shape requires many manipulations and iterations to fine-tune the selection.

### 2.2 Query-by-Content

While this paper attempts to suggest a shape-based brushing technique for trail sets, researchers have explored shape-based selection techniques in different contexts, both using arbitrary shapes and sketch-based queries.

Sketch-based querying presents several advantages over traditional selection [18]. It has been used for volumetric data sets [45], and neural pathway selection [1]. This last work is the closest to the current study. However, the authors presented a domain-specific application and they based their algorithm on the Euclidean distance. This is not a robust metric for similarity detection since it is hard to provide a value indicating a high similarity and varies greatly according to the domain and the data considered. In addition, this metric does not support direction and orientation matching nor the combination of brushing with filtering.

In addition, user-sketched pattern matching plays an important role in searching and localizing time-series patterns of interest [29, 43]. For example, Holz and Feiner [29] defined a relaxed selection technique in which the users draw a query to select the relevant part of a displayed time-series. Correl et al. [18] propose a sketch-based query system to retrieve time-series using dynamic time wrapping, mean square error or the Hough transform. They present all matches individually in a small multiple, arranged according to the similarity measurement. These techniques, as the one proposed here, also take advantage of sketches to manipulate data. However, they are designed for querying rather than selecting, and specifically for 2D data. Other approaches use boxes and spheres to specify the regions of interest [2, 28, 44], and the desired trails are obtained if they intersect these regions of interest. However, many parameters must be changed by the analysts in order to achieve a simple single selection. The regions of interest must be re-scaled appropriately, and then re-positioned back and forth multiple times for each operation. Additionally, many selection box modifications are required to refine the selection and thus hinder and alter the selection efficiency [2].

### 2.3 Similarity measures

Given a set of trajectories, we are interested in retrieving the most similar subset of trajectories with a user-sketched query. Common approaches include selecting the K-nearest-neighbors (KNN) based on the Euclidean distance (ED) or elastic matching metrics (e.g, Dynamic Time Warping - DTWs). Affinity cues have also been used to group objects. For example, objects of identical color are given a high similarity coefficient for the color affinity [40].

The Euclidean distance is the most simple to calculate, but, unlike mathematical *similarity* measurements [59] which are usually bound between 0 and 1 or -1 and 1, ED is unbounded and task-specific. A number of works have suggested transforming the raw data into a lower-dimensional representation (e.g., SAX [39,41], PAA [38, 42]). However, they require adjusting many abstract parameters which are dataset-dependant and thus reduce their flexibility. Lindlbauer presented global and local proximity of 2D sketches. The second measure is used for similarity detection where an object is contained within another one, and is not relevant to this work. While the first measure refers to the distance between two objects (mostly circles and lines), there is no guarantee that the approach could be generalized to large datasets such as eye tracking, GPS or aircraft trajectories. In contrast, Dynamic Time Warping has been considered as the best measurement [20] in various applications [47] to select shapes by matching their representation [24]. It has been used in gestures recognition [58], eye movements [3, 26] and shapes [61]. An overview of existing metrics is available [47].

The k-Nearest Neighbor (*KNN*) approach has also long been studied for trail similarity detection [16, 54]. However, using this metric, two trails may provide a good accurate connection (i.e, a small difference measure as above) even if they have very different shapes. Other measurements to calculate trajectory segment similarity are the Minimum Bounding Rectangles (MBR) [34] or Fréchet Distance [19] which leverage the perpendicular distance, the parallel distance and the angular distance in order to compute the distance between two trajectories.

In order to address the aforementioned issues, we propose and investigate two different approaches. The first approach is based on directly calculating the correlations on *x*-axis and *y*-axis independently between the shape of the brush and the trails (section 3.1.1). The second approach (section 3.1.2) is based on the geometrical information of the trails, i.e, the trails are transformed into a new space (using eigenvectors of the co-variance matrix) which is more suitable for similarity detection. This paper's approach leverages the potential of these two metrics to foster efficient shape-based brushing for large cluttered datasets. As such, it allows targeting detailed motif discovery performed interactively.

## 3 INTERACTION PIPELINE

This section presents the interactive pipeline (Fig. 2) which fulfills the identified design requirements (DR1–DR4). As for any interactive system, user input plays the main role and will operate at every stage of the data processing. First, the input data (i.e. trail set) is given. Next, the user inputs a brush where the pipeline extracts the brushed items, the brush area and its shape. Then, two comparison metrics are computed between every brushed item and the shape of the brush (similarity measurement). A binning process serves to filter the data which is then presented to the user. The user can then refine the brush area and choose another comparison metric until the desired items are selected.

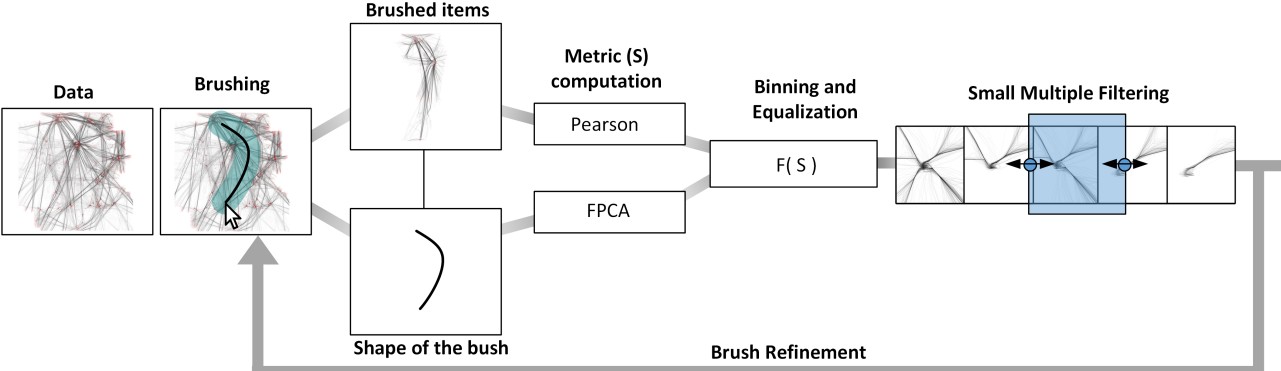

Figure 2: This figure shows the interaction pipeline. The pipeline extracts the brushed items but also the shape of the brush which will be then used as a comparison with the brushed items (metrics stage). Then, brushed items are stored in bins displayed with small multiples where the user can interactivity refine the selection (i.e. binning and filtering stage). Finally, the user can adjust the selection with additional brushing interactions. Note that both the PC and FPCA can be used for the binning process, but separately. Each small multiple comes from one metric exclusively.

## 3.1 Metrics

As previously detailed in the related word section, many comparison metrics exist. While our pipeline can use any metric that fulfills the design requirement DR2 (continuous, ordered and meaningful comparison), the presented pipeline contains only two complementary algorithms: Pearson and FPCA. The first one focuses on a shape comparison basis with correlation between their representative vertices, while the latter focuses on curvature comparison. As shown in Fig. 3, each metric produces different results. The user can use either of them depending on the type of filtering to be performed. During the initial development of this technique, we first considered using the Euclidean distance (ED) and DTW, but we have rapidly observed their limitations and we argue that PC and FPCA are more suitable to trajectory datasets. First, PC values are easier to threshold. A PC value $> 0.8$ provides a clear indication of the similarity of 2 shapes. Moreover, to accurately discriminate between complex trajectories, we need to go beyond the performance of ED. Furthermore, the direction of the trajectories, while being essential for our brushing technique, is not supported by ED and DTW similarity measures. Another disadvantage of using ED is domain- and task-specific threshold that can drastically vary depending on the context. PC, that we used in our approach, on the other hand, uses the same threshold independently of the type of datasets. The two following sections detail the two proposed algorithms.

### 3.1.1 Pearson's Correlation (PC)

Pearson's Correlation (PC) is a statistical tool that measures the correlation between two datasets and produces a continuous measurement between $\in [-1, 1]$ with 1 indicating a high degree of similarity, and $-1$ an anti-correlation indicating an opposite trend [35]. This metric is well suited (DR2) for measuring dataset similarity (i.e. in trajectory points) [17, 55].

Pearson's Correlation $PC$ between two trails $T_i$ and $T_j$ on the $x-axis$ can be defined as follows:

$$r_x = \frac{COV(T_{i_x}, T_{j_x})}{\sigma_{T_{i_x}} \sigma_{T_{j_x}}}, \quad COV(T_{i_x}, T_{j_x}) = E[(T_{i_x} - \overline{T_{i_x}})(T_{j_x} - \overline{T_{j_x}})] \quad (1)$$

Where $\overline{T_{i_x}}$ and $\overline{T_{j_x}}$ are the means, $E$ the expectation and $\sigma_{T_{i_x}}$, $\sigma_{T_{j_x}}$ the standard deviations. The correlation is computed on the $y-axis$ and the $x-axis$ for two-dimensional points.

This metric is invariant in point translation and trajectory scale but it does not take into account the order of points along a

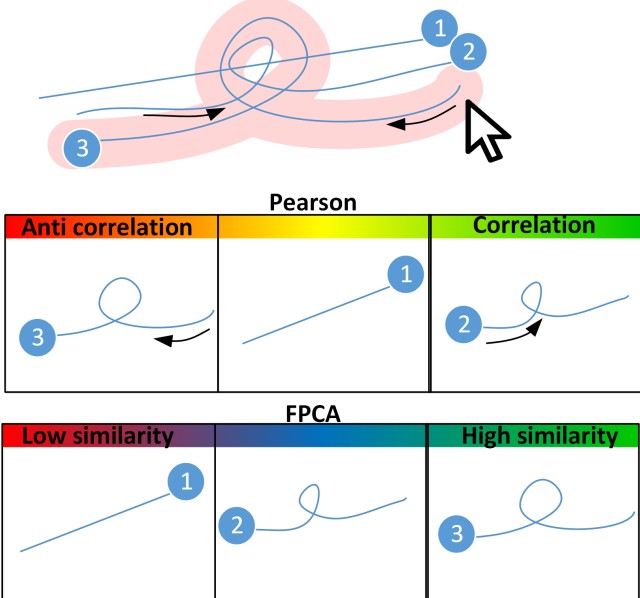

Figure 3: This figure shows an example of the two metrics for shape comparison usage. The user brushed around curve 3 and thus also selected curves 1 and 2. Thanks to the Pearson computation, the associated small multiples show that only curve 2 is correlated to the shape of the brush. Curve 3 is anti correlated since it in an opposite direction to the shape of the brush. The FPCA computation does not take into account the direction but rather the curvature similarity. As such, only shape 3 is considered as highly similar to the brush shape input.

trajectory. Therefore, the pipeline also considers the FPCA metric that is more appropriate to trajectory shape but that does not take into account negative correlation.

### 3.1.2 Functional Principal Component Analysis

Functional Data Analysis is a well-known information geometry approach [48] that captures the statistical properties of multivariate data functions, such as curves modeled as a point in an infinite-dimensional space (usually the $L^2$ space of square integrable functions [48]). The Functional Principal Component Analysis (FPCA) computes the data variability around the mean curve of a cluster

while estimating the Karhunen-Loeve expansion scores. A simple analogy can be drawn with the Principal Component Analysis (PCA) algorithm where eigen vectors and their eigen values are computed, the FCPA performs the same operations with eigen functions (piece-wise splices) and their principal component scores to model the statistical properties of a considered cluster [30]:

$$\Gamma(t, \omega) = \bar{\gamma} + \sum_{j=1}^{+\infty} b_j(\omega)\phi_j(t) \tag{2}$$

where $b_j$ are real-valued random variables called principal component scores. $\phi_j$ are the principal component functions, which obey to:

$$\int_0^1 \hat{H}(s,t)\phi_j(s)ds = \lambda_j\phi_j(t) \tag{3}$$

$\phi_j$ are the (vector-valued) eigenfunctions of the covariance operator with eigenvalues $\lambda_j$. We refer the prospective reader to the work of Hurter et al. [30] for a Discrete implementation. With this model, knowing the mean curve $\bar{\gamma}$ and the principal component functions $\phi_j$, a group of curves can be described and reconstructed (Inverse FPCA) with the matrix of the principal component score $b_j$ of each curve. Usually, a finite vector (with fixed dimension d) of $b_j$ scores is selected such that the explained variance is more than a defined percentile.

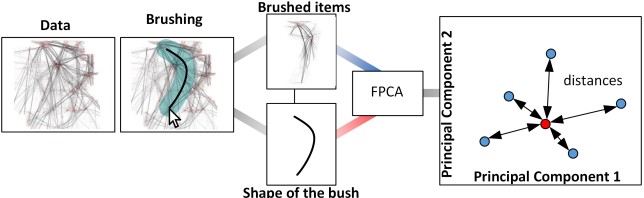

Figure 4: Illustration of FPCA metric algorithm. Diagram of the transformation from the trail space to the 2D points space using FPCA and then measuring the closest points (most similar) in the Principal Component space.

To compute a continuous and meaningful metric (DR2), the metric computation uses the two first Principal Components (PC) to define the representative point of a considered trajectory. Then, the metric is computed by the euclidean distance between the shape of the brush and each brushed trajectory in the Cartesian scatterplot PC1/PC2 (Fig. 4). Each distance is then normalized between $[0, 1]$ with 1 corresponding to the largest difference in shape between the considered shape of the brush of the corresponding trajectory.

### 3.2 Binning and small multiple filtering

Taking into account the computed comparison metrics, the pipeline stores the resulting values into bins. Items can then be sorted in continuous ways from the less similar to the most similar ones. While the Pearson measurements $\in [-1, 1]$ and the FPCA $\in [0, 1]$, this binning process operates in the same way. Each bin is then used to visually show the trajectories it contains through small multiples (we use 5 small multiples which gives a good compromise between visualization compactness and trajectory visibility). The user can then interactively filter the selected items (DR4) with a range slider on top of the small multiple visualizations. The user is thus able to decide whether to remove uncorrelated items or refine the correlated one with a more restricted criterion (DR3).

### 4 INTERACTION PARADIGM BY EXAMPLE

This technique is designed to enable flexible and rapid brushing of trajectories, by both the location and the shape of the brush. The technique's interaction paradigm is now described and illustrated in a scenario where an air traffic management expert studies the flight data depicted in Fig. 6.

### 4.1 Scenario Introduction

Aircraft trajectories can be visually represented as connected line segments that form a path on a map. Given the flight level (altitude) of the aircraft, the trajectories can be presented in 3D and visualized by varying their appearances [7] or changing their representation to basic geometry types [11]. Since the visualization considers a large number of trajectories that compete for the visual space, these visualizations often present occlusion and visual clutter issues, rendering exploration difficult. Edge bundling techniques [37] have been used to reduce clutter and occlusion but they come at the cost of distorting the trajectory shapes which might not always be desirable.

Analysts need to explore this kind of datasets in order to perform diverse tasks. Some of these tasks compare expected aircraft trajectories with the actual trajectories. Other tasks detect unexpected patterns and perform out traffic analysis in complex areas with dense traffic [7, 32]. To this end, various trajectory properties such as aircraft direction, flight level and shape are examined. However, most systems only support selection techniques that rely on starting and end points, or predefined regions. We argue that the interactive shape brush technique would be helpful for these kinds of tasks, as they require the visual inspection of the data, the detection of the specific patterns and then their selection for further examination. As these specific patterns might differ from the rest of the data precisely because of their shape, a technique that enables their selection through this characteristic will make their manipulation easier, as detailed in the example scenario. We consider a dataset that includes 4320 aircraft trajectories of variable lengths from one day of flight traffic over the French airspace.

### 4.2 Brushing

We define the *trail T* as a set of real-valued consecutive points $T = [(Tx_1, Ty_1))^\top, (Tx_2, Ty_2)^\top, \ldots, (Tx_n, Ty_n)^\top]$ where $n$ is the number of points and $(Tx_i, Ty_i)^\top$ corresponds to the $i-th$ coordinate of the *trail*. The Fig. 6 depicts an example of 4133 *trails* (aircraft in French airspace). The brush *Shape* consists of a set of real-valued consecutive points $S = [(Sx_1, Sy_1))^\top, (Sx_2, Sy_2)^\top, \ldots, (Sx_m, Sy_m)^\top]$ where $m$ is the number of points. Note that while the length $n$ of each trail is fixed, the length $m$ of the *Shape* depends on the length of the user brush. The *Shape* is smoothed using a 1€filter [14] and then resampled to facilitate the trail comparison. The similarity metrics are then used in subsequences of the *shape* of approximately the same length as the brush *Shape*. In order to do this, each trail is first resampled so that each pair of consecutive vertices on the trail has the same distance $l_{vertices}$ [22].

The user starts by exploring the data using pan and zoom operations. They are interested in the trajectories from the south-east of France to Paris. The user can choose if they are looking for a subsequence match or an exact match. A subsequence match involves the detection of trajectories having a subsequence similar to the *Shape* locally. Exact match comparison also takes into account the length of the trajectory and the *Shape* in order to select a trajectory, i.e, its length must be approximately similar to the length of the *S*hape (general measurements). This option is especially useful to select a trajectory by its start and end points (e.g, finding trajectories taking off from an airport A and landing at an airport B). The exact matching is supported by analyzing the length of the trail and the *S*hape before applying the similarity metric algorithm.

The analyst in the scenario activates the subsequence match where the Pearson's Correlation metric is selected by default and starts brushing in the vicinity of the target trajectories following the trajectory shape with the mouse. This will define both (1) the brush region and (2) the brush shape, that captures also the brush direction. Once the brushing has been completed, the selected trajectories are highlighted in green, as depicted in Fig. 5-(b).

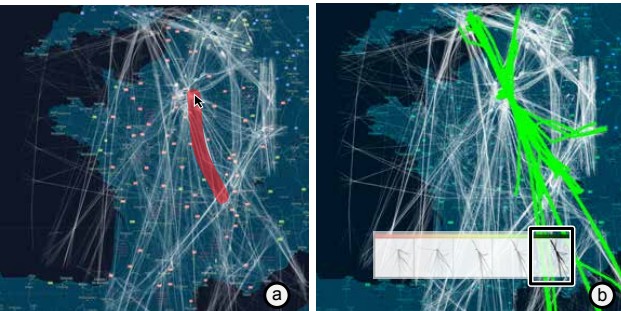

Figure 5: (a) The user brushes the trajectories in order to select those from the south-east of France to Paris. (b) They select the most correlated value to take into account the direction of the trails.

## 4.3 Small Multiples and User filtering

The similarity calculation between the *S*hape and the brushed *region* will produce a *similarity* value for each trail contained in the region and the trails are distributed in small multiples as detailed in Section 3. Once the user has brushed, she can adjust the selection by selecting one of the bins displayed in the small multiples and using its range slider. The range slider position controls the similarity level, and its size determines the number of trajectories selected at each slider position: the smallest size selects one trajectory at a time. The range slider size and position are adjusted by direct manipulation using the mouse. This enables a fine control over the final selection and makes the algorithm thresholding easier to understand as the user controls both the granularity of the exploration and the chosen similarity level. As the bins are equally sized, the distribution of the similarity might not be linear across the small multiples. This makes navigation easier since the trajectories distribution in the small multiples is continuous. However, this also entails that not every bin corresponds to the same similarity value interval. To keep this information available to the user, a colored heatmap (from red to green) displays the actual distribution, as depicted in Fig. 3.

In the current scenario, the expert, as they wish to select only the flights *to* Paris and not *from* Paris, selects the trajectories that are correlated with the original brush, as the correlation takes into account the brush direction. These trajectories are on the right side of the small multiple, highlighted in green as depicted in Fig. 5-(b).

The expert is then interested in exploring the flights that land on the north landing strip but that are not coming from the east. For this, they perform a new shape brush that will consider only the previously selected trajectories to identify the planes that do come from the east, and distinguishable by the "C" shape in the trajectories, as depicted in Fig. 7. To be able to select the geometry precisely, the expert changes to the FPCA metric, using the keyboard shortcut. In this case, the small multiple arranges the trajectories from less similar to more similar. This entails that the small multiple based on FPCA also enables the selection of all the trajectories that *do not* match the specified *S*hape but which are

contained in the brushing region. As all trajectories passing through the north landing strip are contained in the brushing region, the most similar trajectories will correspond to the ones that have a "C" shape, in the same orientation as the *S*hape, and thus come from the east. The less similar will be the ones that interest the analyst, so they can select them by choosing the most dissimilar small multiple as depicted in Fig. 7-(b).

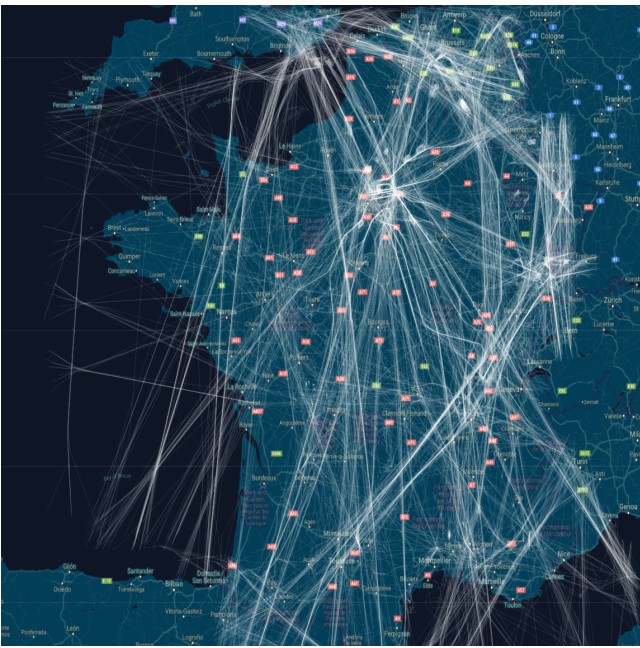

Figure 6: One day's aircraft trajectories in French airspace, including taxiing, taking-off, cruise, final approach and landing phases. Selecting specific trajectories using the standard brushing technique will yield inaccurate results due to the large number of trajectories, occlusions, and closeness in the spatial representation.

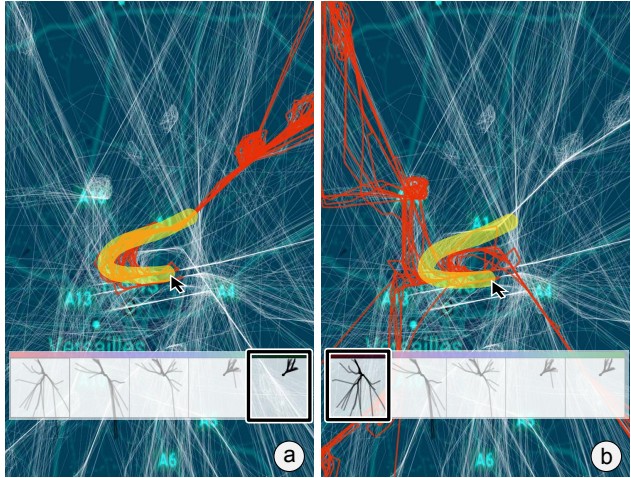

Figure 7: (a) The user filters the trajectories that land on the north runway in Paris by brushing following the "C" shape. This retrieves the flights that come from the east. (b) They change the selection on the small multiples, to select all the dissimilar *S*hapes, resulting in the trajectories that land on the north landing strip but that do not come from the east.

## 5 USE CASES

We argue that there is a strong demand for targeted brushing to select motifs in datasets. In various domains, including aircraft trajectories, eye tracking, GPS trajectories or brain fiber analysis, there is a substantial need to be able to discover hidden motifs in large datasets. Undoubtedly, retrieving desired trails in such datasets would help analysts to focus on the most interesting parts of the data. The system was built using C# and OpenTK on a 64bit [1] XPS 15 Dell Laptop. Although, both PC and FPCA provide different but valuable results, the running performance was 10 times faster with PC compared to FPCA.

The technique was first tested informally with experts from aerospace domain with more than 10 years of experience in trajectories analysis. While the collected feedback was largely positive, we observed some limitations regarding the misunderstanding of our filtering parameters. Given the novelty of our interaction technique, users needed a small training period to better understand the semantic of our small multiples interface. Nevertheless, experts founded our technique interesting and useful since it provides initial good selection result without any parameter adjustment.

Because the presented technique is not designed to replace standard brushing but rather to complement it, we extend the informal user study with an evaluation based on real use cases. We argue that these use cases show how our technique facilitates trajectories selection in dense areas, where standard brushing would require multiple user actions (panning, zooming, brushing).

### 5.1 Eye-tracking Data

Eye-tracking technologies are gaining popularity for analyzing human behaviour, in visualization analysis, human factors, human-computer interaction, neuroscience, psychology and training. The principle consists in finding the likely objects of interest by tracking the movements of the user's eyes [4]. Using a camera, the pupil center position is detected and the gaze, i.e, the point in the scene the user is fixating on, is computed using a prior calibration procedure [15, 25, 27, 49]. Therefore, the gaze data consist of sampled trails representing the movements of the user's eye gaze while completing a given task.

Two important types of recorded movements characterize eye behaviour: the fixations and saccades [21]. Fixations are the eye positions the user fixates for a certain amount of time, in other words, they describe the locations that captured the attention of the user. The saccades connect the different fixations, i.e, they represent the rapid movements of the eye from one location to another. The combination of these eye movements is called the scanpath (Fig. 8A). The scanpath is subject to overplotting. This challenge may be addressed through precise brushing techniques to select specific trails. Usually, fixation events are studied to create an attention map which shows the salient elements in the scene. The salient elements are located at high-density fixation areas. However, the temporal connections of the different fixations provide additional information. The saccades enable the links between the fixations to be maintained and the temporal meaning of the eye movement to be held. Discovering patterns in the raw scanpath data is difficult since, in contrast to aircraft trajectories, eye movements are sparser and less regular (Fig. 8). To address this, different kinds of visualizations for scanpaths have been proposed in the literature. For example, edge bundling techniques [31] minimize visual clutter of large and occluded graphs. However, these techniques either alter trail properties such as shape and

geometric information, or are otherwise computationally expensive, which makes them unsuitable for precise exploration and mining of large trail datasets. Moreover, it is possible to animate eye movements in order to have an insight of the different fixations and saccades. However, given the large datasets of eye movements retrieved from lengthy experiments containing thousands of saccades, this approach is unnecessarily time-consuming and expensive.

Therefore, we next describe how this study's approach supports proper and more efficient motif discovery on such eye-tracking datasets. The tested dataset is adapted from Peysakhovich et al. [46], where a continuous recording of eye movement in a cockpit was performed. The gaze data was recorded at 50 Hz. Sequential points located in a square of $20 \times 20$ pixels and separated by at least 200 ms were stored as a fixation event and replaced by their average in order to reduce noise coming from the microsaccades and the tracking device.

In order to illustrate some examples, we could consider a domain expert who wishes to explore the movements of the pilot's eyes in a cockpit. When performing a task, the pilot scans the different instruments in the cockpit, focuses more on certain instruments or interacts with them. Especially in this context, the order of pilot attention is important since checking a parameter in one instrument may give an indication of the information displayed in another instrument. For example, the priority of the Primary Flight Display (PFD) instrument compared to Flight Control Unit (FCU) will differ for the cruise phase as compared to the final landing approach [46]. As an example of analysis, the user wishes to explore the movement of the eye from the Primary Flight Display (PFD) to the Navigation Display (ND). Selecting these scanpaths using traditional brushing techniques would be challenging because of the clutter, selecting those scanpaths would introduce additional accidental selections. Therefore, he brushes these scanpaths using a shape that extends from the PFD to the ND, applying the Pearson metric to consider the direction. Fig. 8(a) depicts the brushed eye movements that correspond to the most correlated trails in the small multiple. There are several saccades between those two devices, and this is in line with the fact that saccadic movements between the PFD and the ND are typically caused by parameter checking routines.

However, when the user changes the selection and brushes the scanpath between the ND and the FCU, it is surprising to see that there is only one saccade between them. Brushing now with a shape that goes between the PFD and the FCU (Fig. 8-(c)) reveals only one scanpath. This is difficult to visualize in the raw data or using the standard brushing technique. A final *S*hape searching for an eye movement from the PFD to the LAI and passing by the FCU, results in only one saccade (Fig. 8-(d)). To determine the meaning of this behavior, the tool also enables the expert to exploit a continuous transition to increase the visibility and gain insight on when these saccadic movements occurred (temporal view). The user can change the visual mapping from the (x,y) gaze location to the (time,y) temporal view. This smooth transition avoids abrupt change to the visualization [33] (Fig. 9).

### 5.2 GPS Data

GPS trajectories consist of sequential spatial locations recorded by a measurement instrument. Subjects such as people, wheeled vehicles, transportation modes and devices may be tracked by analyzing the spatial positions provided by these instruments. Analysts may need to explore and analyze different paths followed by the users. The advances in position-acquisition and ubiquitous devices have granted extremely large location data, which indicate the mobility of different moving targets such as autonomous vehicles, pedestrians,

---

[1] Intel(R) Core(TM) I7-4712HQ CPU @ 2.30GHz,2301 MHz, 4 core, 8 threads

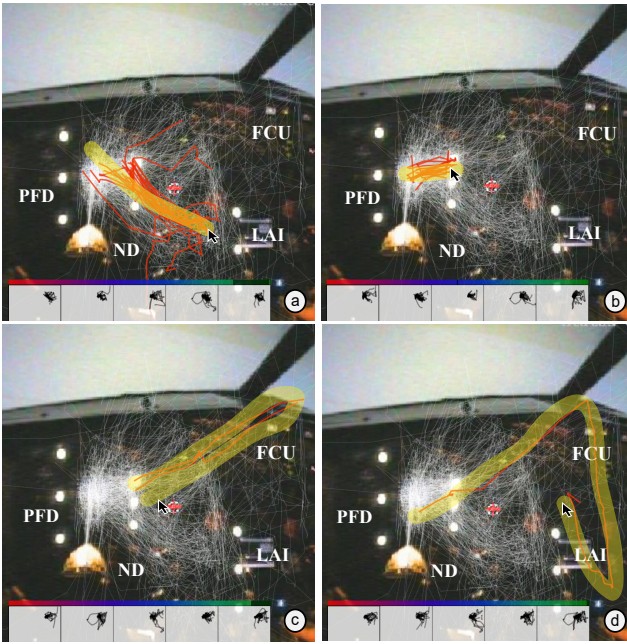

Figure 8: (a) Selected eye movements between the PFD and ND, (b) Selected eye movements in the vicinity of the PFD, (c) Saccades between the ND and the FCU, (d) Eye movement from the PFD to the LAI passing by the FCU.

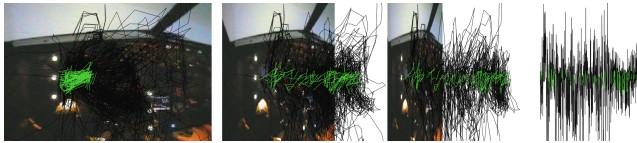

Figure 9: This figure shows the animated transition between the X/Y gaze view to the temporal view. This helps to detected how the selected eye movement occurred over time.

natural phenomena, etc. The commonness of these datasets calls for novel approaches in order to discover information and mine the data [62].

Traditionally, researchers analyse GPS logs by defining a distance function (e.g, *KNN*) between two trajectories and then applying expensive processing algorithms to address the similarity detection. For example, they first convert the trajectories into a set of road segments by leveraging map-matching algorithms. Afterwards, the relationship between trajectories is managed using indexing structures [56, 62]. Using the data provided by Zheng et al. [63], we seek to investigate different locations followed by the users in Beijing. The data consists of GPS trajectories collected for the Geolife project by 182 users during a period of over five years (from April 2007 to August 2012) [63]. Each trajectory is represented by a 3D latitude, longitude and altitude point. A range of users' outdoor movements were recorded, including life routines such as travelling to work, sports, shopping, etc.

As the quantity of GPS data is becoming increasingly large and complex, proper brushing is challenging. Using bounding boxes somewhat alleviate this difficulty by setting the key of interest on the major corners. However, many boxes must be placed carefully for one single selection. The boxes can help the analysts to select all the trajectories that pass through a specific location, but do not

simplify the analysis of overlapping and directional trajectories. This study's approach intuitively supports path differentiation for both overlapping trajectories and takes direction into account. For example, we are interested in answering questions about the activities people perform and their sequential order [63]. For this dataset, the authors were interested in finding event sequences that could inform tourists. The shape-based brushing could serve as a tool to further explore their results. For example, if they find an interesting *classical sequence* that passes through locations A and B they can further explore if this sequence corresponds to a larger sequence and what other locations are visited before or after. A first brushing and refinement using the FPCA metric and small multiples enables them to select all the trajectories that include a precise *event sequence* passing through a set of locations, as depicted in Fig. 10. A second brushing using the Pearson metric enables further explorations that also take into account the direction of the trajectories. Switching between the correlated trajectories and the anti-correlated ones, the user can gain insight about the visitation order of the selected locations.

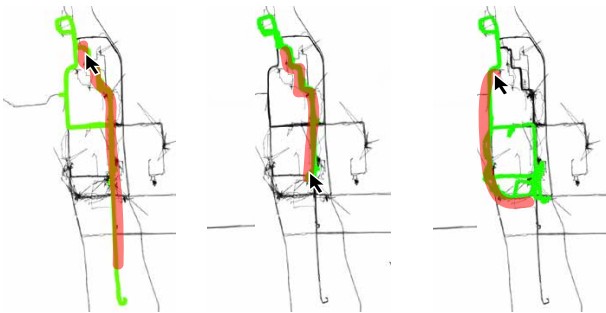

Figure 10: GPS locations of pedestrians (black). Selection of three different trajectories containing three different event sequences from [63] (green).

## 6 DISCUSSION

The proposed brushing technique leverages existing methods with the novel usage of the shape of the brush as an additional filtering parameter. The interaction pipeline shows different data processing steps where the comparison algorithm between the brushed items and the shape of the brush plays a central role. While the presented pipeline contains two specific and complementary comparison metric computations, another one can be used as long as it fulfills the continuity and metric semantic requirements (DR2). There are indeed many standard approaches (ED, DTW, Discrete Fréchet distance) that are largely used by the community and could be used to extend our technique when faced with different datasets. Furthermore, the contribution of this paper is a novel shape-based brushing technique and not simply a shape similarity measure. In our work, we found two reasonable similarity measures that fulfill our shape-based brushing method: The FPCA distance comparison provides an accurate curve similarity measurement while the Pearson metric provides a complementary criteria with the direction of the trajectory.

In terms of visualization, the binning process provides a valuable overview of the order of the trajectory shapes. This important step eases the filtering and adjustment of the selected items. It is important to mention that this filtering operates in a continuous manner as such trajectories are added or removed one by one when adjusting this filtering parameter. This practice helps to fine tune the selected items with accurate filtering parameters. The presented scenario shows how small multiple interaction can provide flexibility. This is especially the case when the user

brushes specific trajectories to be then removed when setting the compatibility metrics to uncorrelated. This operation performs a brush removal. The proposed filtering method can also consider other types of binning and allows different possible representations (i.e. various visual mapping solutions).

This paper illustrates the shape-based brushing technique with three application domains (air traffic, eye tracking, GPS data), but it can be extended to any moving object dataset. However, our evaluation is limited by the number of studied application domains. Furthermore, even if various users and practitioners participated in the design of the technique, and assessed the simplicity and intuitiveness of the method, we did not conduct a more formal evaluation. The shape-based brush is aimed at complementing the traditional brush, and in no way do we argue that it is more efficient or effective than the original technique for all cases. The scenarios are examples of how this technique enables the selection of trails that would be otherwise difficult to manipulate, and how the usage of the brush area and its shape to perform comparison opens novel brushing perspectives. We believe they provide strong evidence of the potential of such a technique.

The technique also presents limitations in its selection flexibility, as it is not yet possible to combine selections. Many extensions can be applied to the last step of the pipeline to support this. This step mainly addresses the DR4 where the selection can be refined thanks to user inputs. As such, multiple selections can be envisaged and finally be composed. Boolean operations can be considered with the standard And, Or, Not. While this composition is easy to model, it remains difficult for an end user to master the operations when there are more than 2 subset operations [33, 60]. As a solution, Hurter et al. proposed an implicit item composition with a simple drag and drop technique [33]. The pipeline can be extended with the same paradigm where a place holder can store filtered items and then be composed to produce the final result. The user can then refine the selection by adding, removing or merging multiple selections.

## 7 CONCLUSION

In this paper, a novel sketch-based brushing technique for trail selection was proposed and investigated. This approach facilitates user selection in occluded and cluttered data visualization where the selection is performed on a standard brush basis while taking into account the shape of the brush area as a filtering tool. This brushing tool works as follows. Firstly, the user brushes the trajectory of interest trying to follow its shape as closely as possible. Then the system pre-selects every trajectory which touches the brush area. Next, the algorithm computes a distance between every brushed shape and the shape of the brushed area. Comparison scores are then sorted and the system displays visual bins presenting trajectories from the lowest scores (unrelated - or dissimilar trajectories) to the highest values/scores (highly correlated or similar trajectories). The user can then adjust a filtering parameter to refine the actual selected trajectories that touch the brushed area and which have a suitable correlation with the shape of the brushed area. The cornerstone of this shape-based technique relies on the shape comparison method. Therefore, we choose two algorithms which provide enough flexibility to adjust the set of selected trajectories. One algorithm relies on functional decomposition analysis which ensures a shape curvature comparison, while the other method insures an accurate geometric based comparison (Pearson algorithm). To validate the efficiency of this method, we show three examples of usage with various types of trail datasets.
This work can be extended in many directions. We can first extend it with additional application domains and other types of dataset such as car or animal movements or any type of time-varying data. We

can also consider other types of input to extend the mouse pointer usage. Virtual Reality data exploration with the so-called immersive analytic domain gives a relevant work extension which will be investigated in the near future. Finally, we can also consider adding machine learning to help users brush relevant trajectories. For instance, in a very dense area, where the relevant trajectories or even a part of the trajectories are not visible due to the occlusion, additional visual processing may be useful to guide the user during the brushing process.

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
