# OpenReview forum: "Interactive Shape Based Brushing Technique for Trail Sets"
_graphicsinterface.org/Graphics_Interface/2020/Conference — GI 2020_

### Official Review · AnonReviewer1 · 2020-01-08
**Interesting idea and implementation, but lacks maturity**

**Confidence:** 4
**Rating:** 5

**Review:**

This paper presents a technique for brushing trajectories based on two different metrics, and provides case studies illustrating the efficiency of the technique.


I am not an expert in brushing per se, but I find the related work accessible enough for a non-expert to grasp the research landscape in the area. The authors do a great job at identifying limitations of related work to position their contribution with clarity.
One thing I find lacking in the introduction, however, is the motivation for this work. It comes through later, but should be clearer upfront. Why is brushing 3D trajectories (in large datasets) important? In which domain/context is it useful? What are the current techniques being used and what are their limitations? Clarifying these points (possibly through an example) would strengthen the argument. Along these lines, the requirements make sense to me, but they seem rather arbitrary too. Tying them to the motivation and related work/background will also make a stronger case.

The proposed algorithms make sense. I have a few comments though:
- why using Pearson in 2D while the focus of the paper is supposed to be on 3D trajectories?
- The FCPA section is difficult to read and would benefit from a rewriting. The figures help though.
- Did the authors consider establishing a metric that would be a weighted mean of the two metrics? This would be worth mentioning.

The binning and small multiple filtering is not very well explained. It i difficult to grasp how it works. How are the 5 small multiples selected, when the two metrics are on different scales? How does one selects, filters, or adjusts one of the brushes? Is it possible to weigh the different small multiples? Answers to these questions come later in the scenario, but make the explanation confusing at first. I recommend moving these explanations before the scenario.s

It is disappointing to read that the authors collected feedback from domain experts, to only ignore this feedback after all. Given that the experts misunderstood the filtering parameters, I would expect the authors to at least iterate over a few alternatives and go back to the experts with those. What I read is that the authors assume that the default parameters are good enough thus they do not worry that domain experts do not understand how to change the parameters; thus decide to not try further to design a system that would allow the experts to benefit from the full power/flexibility of the technique. I have a hard time understanding why that is.
A straightforward improvement could be to explain the metrics and their parameters in understandable terms, rather than technical ones. For example, replacing "Pearson" by "Direction-aware" or something similar would certainly make its purpose clearer.

The example with flight trajectories is quite interesting to read. The case study with eye tracking data, on the other hand, is hard to follow. It is an interesting case study because the approach is clearly different from what is being used currently to analyze gaze data, however, it requires some heavy re-writing.

Another point of concern I have is that the authors state that they "found two reasonable similarity measures that fulfill
our shape-based brushing method", but do not show evidence that they tried others. This is intriguing, because the authors argue that their pipeline is easily adaptable to other metrics, thus I would expect them to implement a bunch of metrics and show a comparison of the results. This would provide much more valuable knowledge (which gives which result for which dataset and which brush) than just reporting the results for two somewhat arbitrarily chosen metrics.

Overall, this work clearly has merits: it is an interesting problem, a good implementation of a reasonable solution, and some interesting discussion of the technique. However, the paper lack clarity and focus throughout. The writing is not great, and some of the arguments are difficult to follow because the information is scattered throughout the paper. There are also some more fundamental issues, like having arbitrary design requirements, ignoring domain expert feedback, and not exploring further metrics although the pipeline was designed with this in mind. So, while the paper is fine in its current state, it would likely become a much better paper after a round of revisions.

Last, the writing is fine but the language could be improved. There are also a few typos or grammatical mistakes, including:
- This figure shows the interaction pipeline to filtered items
- Followed by the binning process and its filtering, the resulting data is presented to the user.
- our technique provides initial good selection result
- Discrete FrÃl’chet distance

There is also a leftover comment in the paper: USE CASES, second paragraph (ML: 3?).

---

### Official Review · AnonReviewer3 · 2020-01-08
**Interesting approach to brushing**

**Confidence:** 5
**Rating:** 6

**Review:**

The authors describe the design and implementation of a shape-based brushing technique targeted at selecting a particular type of data - trajectories. These are notoriously difficult to select directly due to issues of occlusion and the "hairball" effect when there are many trajectories intertwines, as is the case with eye tracking, network, or flight trails data. The authors do an excellent job of describing the problem  and grounding the approach in previous work.  The approach is interesting and the use cases described demonstrate the technique well.   However, the paper is weakened by several writing and organizational aspects, and by an odd off-hand report of user feedback.

The basics of the technique are well-described: the user draws a shape that the system then selects matches for, based on two similarity metrics (one calculated by Pearson's coefficient  and the other by a PCA algorithm). As these two metrics deliver different candidates, the resulting set of trajectories is provided to the user in a set of small multiples illustrating the selected trajectories and sorted by similarity; the user can refine selections, although it was not clear how. There appears to be a set of small multiples for each of the two metrics.
One main weakness of the paper is manifested here: I found the description of the bins, and how they are calculated, quite confusing. I had to re read the paper back and forward to finally tease out what I think is the way it works.  Overall, the writing and the organization of the paper suffered from similar issues.  A similar problem occurred with a critical aspect of the brushing technique: direction. The authors state directionality is a critical advantage of their brushing technique, but never actually stipulate how direction is specified in the original share definition. I assuming - as one would consider the obvious choice - that directionality is taken from the direction of the sketched brush at the time the user draws it. But this is not clear. IN fact, the whole way the user draws the shape is poorly described. The nice video provided was helpful in showing this technique. However, the video alludes to something not mentioned in the paper about directionality: only the Pearson algorithm identifies direction, and even from the video it was not clear how the user selected it.

These critical areas of confusion around how the process actually unfolds from start to finish should have been more clearly described.

I found it odd that at the authors retained both metrics, delivering different results, without trying some blended version that might reduce complexity for the user. One would expect that trying some combination would be an obvious step, especially given the unclear feedback from the expert review.

The last point leads me to what I see as *the* major weakness of the paper. Having reviewed this approach with experts, the authors state that the experts “did not get it”, and so they choose to describe the system with a use-case method. In fact, this reads as if the feedback from the experts was so bad that they did not want to describe it. Why don’t they include the feedback? Surely they found out useful information.   It sounds like a classic case of “there’s nothing wrong with our system, just change the user”.

Because of that last point, I am somewhat on the fence about this paper, but am willing to consider that it might be acceptable. I’d like to see an inclusion of the user review.

---

### Official Review · AnonReviewer2 · 2020-01-10
**Useful technique with many applications**

**Confidence:** 5
**Rating:** 8

**Review:**

This paper presents a technique for sketch-based brushing of multiple trails ("trail sets") to select one or more trails with precision. The technique uses two different measures (FPCA and Pearson) to find trails that are similar in shape or similar in direction to the sketched path. The applications discussed include selecting GPS-driven paths of aircraft or cars, or selecting scanpaths from eye-tracking for further analysis.

While the paper has many minor typos and errors, they did not detract from my understanding, and the work is generally well-written and clear (with one major exception - see next paragraph) and discusses the relevant literature in a thorough and balanced manner. The figures illustrate the technique appropriately, though some of them are low resolution and hard to follow - in particular the small multiples in Figures 5 and 7. Please embed high-resolution figures so an interested reader can zoom in.

The small multiples view and how it works took me a while to figure out. And I'm still not sure how someone would "refine" a query other than redrawing a trajectory or changing the size of an existing brush (and that's really just my guess). The end of the paper suggests that the queries can't be combined, so it seems to be a bit strange to say that query refinement is possible at all. The use case discussion on page 8 bottom left, it says that FPCA and small multiples can be used first to find matching shapes, then Pearson can be used to take into account directionality. Given that the queries cannot be combined to allow for iterative refinement of selections, it was not clear to me how the Pearson measure would help if it cannot match the shape well. If you have to start over again with a new query, isn't all the shape matching lost? Wouldn't it make sense to first refine by shape, then further refine by direction? The way the measures can be used together (and the types of things they capture) should be further clarified with a figure.

Furthermore, the way the range slider is used in the small multiples isn't clear when it is first introduced on page 4. It becomes clearer in the interaction discussion on page 5, but I'm not sure these two sections are needed and I would advise to consider merging them into a single discussion of the small multiples view and how to interact with it. Also, a better depiction of the range slider in use would help. I liked the use of the color scale atop the multiples to clarify the uneven bin sizes that result from the specific distribution of the measure of interest. The video really helped clarify this (though, a narration would be more interesting than the subtitles, if possible).

The technique is not evaluated with any sort of user study or formal analysis, but rather illustrated with some use cases created by the authors. That said, to me the utility is clear and convincing, and I do not see the need for a study other than perhaps to better understand if people can figure out how to use the small multiples appropriately (or if they know when to use each measure). For usability, it may be better to call the similarity modes "shape" and "direction" rather than using their formal names on the interface itself.

Overall, with some editing, I think this would be a good contribution to GI.

The following minor issues should be fixed:
- Remove references to section numbers as there are no numbers in this format
- Order grouped references so the orders appear sequential [13][41] rather than [41][13]
- The rendering of "Shape" is odd and inconsistent on page 5. Sometimes the S is bold, sometimes not. I suggest just use normal font and don't worry about constantly calling out the relation to the S vector of shape points - the relationship is clear.
- Left of page 5 trial should be trail
- The abstract is too long
- Some spacing issues, e.g. space before period on page 1, no space before reference 44 on page 2 (there are more throughout)
- In the final scene of the video the small multiples extend beyond the boundaries of the frames - is that a bug?
- Page 6 "ML 3?" seems to be an editing note
- Figure 8 c it appears to be 2 trails not one that appear between ND and FCU. Also, the caption has spacing issues
- Page 8 top right "Fr???et distance" (looks like a special character error)
- Figure 10 caption needs to be clarified to explain what is meant by different event sequences - that is means the green complete paths have different patterns
-
-

---

### Meta-Review · Area_Chair1 · 2020-01-11

**Recommendation:** Accept
**Confidence:** 4

**Metareview:**

The reviewers were split on this paper, raising significant concerns about the understandability of some aspects, including the query refinement, small multiples, and the way the PCA works. They have made concrete suggestions about sections that need attention and potential reorganization which could improve this manuscript immensely. I suggest the authors pay careful attention to these recommendations as well as the minor edits in order to improve the paper before presentation.

Two reviewers raised concerns about the treatment of the expert feedback (that it was collected, then disregarded). Two reviewers mentioned that the challenges may have come from the usability of a system with names like "Pearson" on the tools rather than more semantically meaningful names, but it's hard to say what the reasons could be as the feedback is not reported at all. We recommend that at least some of the expert feedback (even if negative) be discussed, to help readers know what to take away from the feedback in case they wish to reimplement the reported techniques and build on what was learned from the study.

Overall, the reviews lean weakly towards "accept".

---

### Decision · Program_Chairs · 2020-01-11

Accept